# Proof of Concept Study for Increasing Tenascin-C-Targeted Drug Delivery to Tumors Previously Subjected to Therapy: X-Irradiation Increases Tumor Uptake

**DOI:** 10.3390/cancers12123652

**Published:** 2020-12-05

**Authors:** Aya Sugyo, Atsushi B. Tsuji, Hitomi Sudo, Kanako Takano, Moriaki Kusakabe, Tatsuya Higashi

**Affiliations:** 1Department of Molecular Imaging and Theranostics, National Institute of Radiological Sciences, National Institute for Quantum and Radiological Science and Technology (QST-NIRS), Chiba 263-8555, Japan; sugyo.aya@qst.go.jp (A.S.); sudo.hitomi@qst.go.jp (H.S.); takano.kanako@qst.go.jp (K.T.); higashi.tatsuya@qst.go.jp (T.H.); 2Graduate School of Agricultural and Life Science, The University of Tokyo, Tokyo 113-8654, Japan; amkusa@mail.ecc.u-tokyo.ac.jp; 3Matrix Cell Research Institute Inc., Ushiku 300-1232, Japan

**Keywords:** injured tissue, tissue remodeling, treatment resistance, radioisotope, cancer stroma

## Abstract

**Simple Summary:**

We hypothesized that an agent recognizing a specific factor, which is involved in tissue injury repair, could achieve the goal of delivering an additional antitumor agent to tumors during tissue repair after initial anticancer therapy. To demonstrate our concept, the present study employed tenascin-C (TNC) as a target molecule and radiation as initial therapy. Increased TNC expression was observed in tumors after radiation exposure in a pancreatic cancer mouse model. Of our three anti-TNC antibodies, the antibody 3–6 showed statistically significant higher tumor uptake compared with non-irradiated tumors in the by biodistribution and single-photon emission computed tomography with computed tomography studies. This finding strongly supports our concept. Our proposed therapeutic strategy could result in better outcomes for patients with treatment-refractory cancer.

**Abstract:**

In treatment-refractory cancers, tumor tissues damaged by therapy initiate the repair response; therefore, tumor tissues must be exposed to an additional burden before successful repair. We hypothesized that an agent recognizing a molecule that responds to anticancer treatment-induced tissue injury could deliver an additional antitumor agent including a radionuclide to damaged cancer tissues during repair. We selected the extracellular matrix glycoprotein tenascin-C (TNC) as such a molecule, and three antibodies recognizing human and murine TNC were employed to evaluate X-irradiation-induced changes in TNC uptake by subcutaneous tumors. TNC expression was assessed by immunohistochemical staining of BxPC-3 tumors treated with or without X-irradiation (30 Gy) for 7 days. Antibodies against TNC (3–6, 12–2–7, TDEAR) and a control antibody were radiolabeled with ^111^In and injected into nude mice having BxPC-3 tumors 7 days after X-irradiation, and temporal uptake was monitored for an additional 4 days by biodistribution and single-photon emission computed tomography with computed tomography (SPECT/CT) studies. Intratumoral distribution was analyzed by autoradiography. The immunohistochemical signal for TNC expression was faint in nontreated tumors but increased and expanded with time until day 7 after X-irradiation. Biodistribution studies revealed increased tumor uptake of all three ^111^In-labeled antibodies and the control antibody. However, a statistically significant increase in uptake was evident only for ^111^In-labeled 3–6 (35% injected dose (ID)/g for 30 Gy vs. 15% ID/g for 0 Gy at day 1, *p* < 0.01), whereas limited changes in ^111^In-labeled TDEAR2, 12–2–27, and control antibody were observed (several % ID/g for 0 and 30 Gy). Serial SPECT/CT imaging with ^111^In-labeled 3–6 or control antibody provided consistent results. Autoradiography revealed noticeably stronger signals in irradiated tumors injected with ^111^In-labeled 3–6 compared with each of the nonirradiated tumors and the control antibody. The signals were observed in TNC-expressing stroma. Markedly increased uptake of ^111^In-labeled 3–6 in irradiated tumors supports our concept that an agent, such as an antibody, that recognizes a molecule involved in tissue injury repair, such as TNC, could enhance drug delivery to tumor tissues that have undergone therapy. The combination of antibody 3–6 coupled to a tumoricidal drug and conventional therapy has the potential to achieve better outcomes for patients with refractory cancer.

## 1. Introduction

Continuous advances in cancer therapy have led to improved survival of patients with many types of cancer [1]. Despite such advances, however, the prognosis of patients with a treatment-refractory cancer, as is often the case for pancreatic cancer, remains poor [1,2]. Most patients with refractory cancer receive multimodal therapy consisting of chemotherapy and radiation [3]. Although the outcome of patients with refractory cancer is unpredictable [2,3], anticancer treatments clearly cause damage to cancer tissues, suggesting that the cancer tissues initiate a physiological response to treatment-induced injury. Therefore, we hypothesized that an agent recognizing a molecule associated with that response could also mediate the delivery of anticancer drugs and radionuclides and thereby provide additional therapeutic benefit.

Tenascin-C (TNC) is an extracellular matrix glycoprotein that participates in cell adhesion, growth, migration, and differentiation [4,5,6]. TNC is expressed at a low level in healthy adult tissues, yet it is upregulated substantially and specifically in response to tissue injury [5,6]. The upregulation of TNC plays a role in tissue repair in damaged tissues but also can promote the growth, differentiation, vascularization, cell adhesion, invasion capacity, and metastatic potential of tumors [5,6]. TNC is a hexameric glycoprotein [5] that provides many potential binding sites for anticancer agents such as antibodies. Therefore, TNC is an attractive target molecule for testing our hypothesis that a drug delivery mechanism targeting a tissue injury responsive factor could increase the overall efficacy of an anticancer regimen.

We developed several antibodies against TNC, including three that recognize both human and murine TNC; these antibodies were named 3–6 [7], 12–2–7 [8], and TDEAR2 [8], as shown in Figure 1. As a tumor model, a BxPC-3 pancreatic cancer xenograft tumor model was selected because BxPC-3 tumor tissues produce only small amounts of TNC in control/nontreated animals, yet substantial amounts are produced after X-ray irradiation, so this approach was appropriate to test our hypothesis. The three antibodies were radiolabeled with ^111^In, and changes in the uptake of the radiolabeled antibodies were evaluated in nude mice bearing tumors that had been previously subjected to X-irradiation (or were not irradiated, as a control).

## 2. Results

### 2.1. Analysis of TNC Expression in Tumors Treated with X-Irradiation

Immunohistochemical staining of sections of nonirradiated BxPC-3 tumors revealed only faint TNC intensity in the stroma and none in tumor cells (Figure 2). This lack of expression in tumor cells was confirmed by cell-binding assays, in which there was no binding of ^111^In-labeled antibody 3–6 to BxPC-3 cells in vitro (Figure 3). In tumors exposed to X-rays (30 Gy), the TNC-stained area as well as staining intensity in the stroma increased progressively until day 7 post-exposure (Figure 2). Therefore, we chose day 7 post-irradiation as the starting point for further experimentation with irradiated samples.

### 2.2. Biodistribution of ^111^In-Labeled Antibodies

At 7 days after irradiation of 30 Gy, each individual ^111^In-labeled antibody was injected into mice bearing tumors, and the biodistribution of each antibody was evaluated after 30 min and on days 1, 2, and 4 post-injection (Figure 4). Figure 5 presents data showing the temporal changes of ^111^In-labeled antibody uptake in nonirradiated tumors and those irradiated at 30 Gy. In nonirradiated tumors, uptake of ^111^In-labeled 3–6 and TDEAR2 was greater than that of ^111^In-labeled control antibody (*p* < 0.01), whereas the uptake of ^111^In-labeled 12–2–7 did not differ significantly compared with the control antibody. Although uptake of each of the four antibodies by tumors irradiated with 30 Gy was greater than that of nonirradiated tumors, there were significant differences among tumors of mice injected with 3–6, TDEAR2, or control antibodies (*p* < 0.01 or *p* < 0.05), whereas no significant difference was observed with 12–2–7 (Figure 5). Tumor uptake of ^111^In-labeled 3–6 increased markedly, i.e., 35% injected dose per gram (ID/g), at day 1 post-injection (Figure 5), which was more than 2-fold greater than for nontreated tumors (Figure 5). Table 1, Table 2, Table 3 and Table 4 show the biodistribution of the four ^111^In-labeled antibodies in normal organs. Although there were several statistically significant differences between the nonirradiated and 30 Gy groups for each antibody among the various organs, other differences were marginal (Table 1, Table 2, Table 3 and Table 4). The statistical differences were found in bone uptake of the control antibody (Table 1), in the spleen of 3–6 (Table 2), the pancreas and kidney of 12–2–7 (Table 3), and the liver of TDEAR2 (Table 4). Although the exact reasons are unclear, the different tumor uptake might affect those uptakes.

### 2.3. Single-Photon Emission Computed Tomography and Computed Tomography (SPECT/CT) with ^111^In-Labeled Antibodies

SPECT/CT (single-photon emission computed tomography with computed tomography) imaging of mice injected with ^111^In-labeled control antibody or antibody 3–6 was conducted to confirm the results of the biodistribution study. Figure 6 presents serial SPECT/CT images after 30 min and on days 1, 2, 3, and 4 post-injection of each labeled antibody. At 30 min post-injection, the radioactivity of both ^111^In-labeled antibodies in the blood pool was very high, whereas that in tumors was low. At day 1, the uptake of ^111^In-labeled 3–6 in tumors irradiated with 30 Gy had increased markedly compared with the 30 min time point and was substantially greater than that for the nonirradiated tumor and for tumors of mice injected with the ^111^In-labeled control antibody. Although on day 2 or later, tumor uptake of ^111^In-labeled 3–6 decreased to approximately half, it remained higher compared with the nonirradiated tumor and the control antibody. There was no unexpected high uptake in organs and tissues. These findings are consistent with those in the biodistribution study as mentioned above.

### 2.4. Autoradiography

On day 1 post-injection of ^111^In-labeled antibody 3–6 or control antibody, nonirradiated and 30 Gy–irradiated tumors were excised and sectioned for autoradiography. Only a slight radiation signal was evident in each of the nonirradiated and irradiated tumors from mice injected with the labeled control antibody (Figure 7A). In the irradiated tumors of mice injected with ^111^In-labeled 3–6, the area encompassed by the strong signal was much greater than that for the nonirradiated tumor (Figure 7A). There was no significant difference in signal intensity between 0 Gy and 30 Gy in the control antibody group, while there was a significant difference in the antibody 3–6 group (*p* < 0.01) shown in Figure 7B. The signal intensity for ^111^In-labeled 3–6 was significantly higher than that for the control (*p* < 0.01, Figure 7B). The area with the strong radioactivity signal for the ^111^In-labeled antibody 3–6 also showed intense staining for TNC in adjacent sections, and that with the weak signal showed low staining (Figure 8).

### 2.5. Immunohistochemistry to Compare Antibodies 3–6 and 12–2–7

As a separate experiment, immunohistochemical staining of adjacent sections with antibodies 3–6 and 12–2–7, which bind to the EGF (epidermal growth factor)-like repeats (Figure 1), was carried out. Although both antibodies stained stroma but not tumor cells, the staining pattern of stroma was different; namely, antibody 3–6 stained TNC throughout the stroma, whereas antibody 12–2–7 stained only part of the stroma (Figure 9).

## 3. Discussion

We hypothesized that an antibody recognizing a molecule associated with tissue injury repair after antitumor therapy could deliver an additional tumoricidal agent, such as a radionuclide, to cancer tissues. To test this hypothesis, we selected TNC as a target molecule and employed three antibodies (3–6, 12–2–7, and TDEAR2) recognizing human and murine TNC [7,8]. These antibodies were labeled with ^111^In, and temporal changes of the uptake of each antibody were evaluated in nude mice bearing BxPC-3 tumors exposed to X-rays, which induce TNC expression. The biodistribution studies revealed markedly increased tumor uptake of ^111^In-labeled antibody 3–6 with statistical significance (35% ID/g for 30 Gy vs. 15% ID/g for 0 Gy at day 1, *p* < 0.01). SPECT/CT imaging and autoradiographic studies provided consistent results. These findings demonstrate that an anti-TNC antibody could deliver a radionuclide to tumors, supporting our hypothesis.

Our proposed therapeutic strategy with the anti-TNC antibody coupled with an antitumor agent has three advantages. First, it targets intratumoral regions responding to damage induced by initial cancer therapy as shown in the present study, providing additional burden before successful repair. This strategy could also circumvent the problem of resistance to therapy. Second, the strategy reduces stromal barriers within the tumor microenvironment, as such barriers can inhibit the penetrance of antitumor agents, especially high-molecular-weight agents, into tumors [9]. More intratumoral stroma is formed by anticancer therapy; TNC is induced and plays a role in stroma formation [5,6,10]. Our antibody 3–6 targets upregulated TNC and could inhibit stroma formation. Third, although antibodies will generally accumulate in tumor tissues at a relatively slow rate [11], our antibody 3–6 accumulated rapidly in the tumors that had undergone therapy, indicating that radiolabeled 3–6 can deposit higher radiation doses in tumors. Taken together, the therapeutic strategy with antibody 3–6 conjugated to a tumoricidal agent, including radionuclides, has the potential to provide better outcomes when combined with conventional therapy.

Interestingly, the present study revealed a difference in tumor uptake of the three anti-TNC antibodies 3–6, 12–2–7, and TDEAR2. TDEAR2 recognizes a region in TNC derived from an alternatively spliced pre-mRNA, suggesting that the majority of TNC that is upregulated upon exposure of tumors to X-rays does not contain this region. Previous studies showed that the upregulation of TNC in response to a toxin or hapten yields the splice variants of TNC [7,12]. The splicing of TNC pre-mRNA underlies the observed spatiotemporal expression of TNC, which is associated with distinct cellular processes [13]. Our data suggest that TNC pre-mRNA is perhaps spliced in tumors after X-ray exposure, and to date, no other studies have shown this. Additional studies might provide new insights into the complexity of TNC functions during tissue repair. Although antibodies 3–6 and 12–2–7 recognize EGF-like repeats of TNC [7,8], the staining patterns for these two antibodies differed in cancer tissues, suggesting that the two recognize different epitopes. There are several glycosylation sites in the EGF-like repeats region [5], so this particular post-translational modification might affect the recognition of TNC by 3–6 and 12–2–7, leading to the different rates of uptake of the two antibodies. Further epitope analysis could reveal the reason why antibody 3–6 was taken up more aggressively by injured tumors, enabling optimization of our therapeutic strategy.

Our study has several limitations. First, the stroma of nonirradiated BxPC-3 tumors expressed only a small amount of TNC, whereas TNC is highly expressed in tumor stroma of many epithelial malignancies including pancreatic cancer [8,10]. Therefore, it will be necessary to evaluate changes in the uptake of anti-TNC antibodies in tumors that express a high level of TNC under the untreated condition. Second, upregulation of TNC expression is induced by antitumor drugs as well as by radiation [5,14]. X-rays achieve uniform distribution of radiation in cancer tissues, whereas chemotherapy and nuclear-medicine therapy result in heterogeneous distribution of drugs and radionuclides, respectively. Therefore, it will be necessary to evaluate to what extent tumor uptake changes after chemotherapy and/or radionuclide therapy to clarify what types of therapy could be combined with our proposed antibody-mediated treatment strategy.

In conclusion, the present study demonstrates that antibody 3–6 can deliver a radionuclide additionally to BxPC-3 tumors previously exposed to X-rays. This supports our concept that an antibody recognizing a specific factor, such as TNC, which is involved in tissue injury repair, could achieve the goal of delivering an additional antitumor agent to tumors during tissue repair after initial cancer therapy. A combination of antibody 3–6 with conventional cancer therapy could result in better outcomes for patients with treatment-refractory cancer.

## 4. Materials and Methods

### 4.1. Antibody

Rat antibodies 3–6 [7], 12–2–7 [8], and TDEAR2 [8] that recognize human and murine TNC were developed previously, and their epitopes are shown in Figure 1. As a control, IgG from rat serum was obtained from Sigma (St. Louis, MO, USA).

### 4.2. Cells

The human pancreatic cancer-cell line BxPC-3 and the human melanoma-cell line A374 were obtained from ATCC (Manassas, VA, USA). The cells were maintained in RPMI1640 medium (Wako Pure Chemical Industries, Osaka, Japan) supplemented with 10% fetal bovine serum (Sigma) in a humidified incubator maintained at 37 °C with 5% CO_2_.

### 4.3. Cell Binding of ^111^In-Labeled Anti-TNC 3–6

The binding of ^111^In-labeled anti-TNC 3–6 to BxPC-3 cells was carried out as previously described [15]. Briefly, 3–4 days after seeding, BxPC-3 cells were detached and suspended in phosphate-buffered saline with 1% BSA (Sigma, St. Louis, MO, USA) at various densities ranging from 3.9 × 10^4^ to 1.0 × 10^7^ (*n* = 3 per number of cells). Each suspension was incubated with ^111^In-labeled anti-tenascin-C (TNC) 3–6 on ice for 60 min. After washing the cells, radioactivity bound to cells was measured using a gamma counter.

### 4.4. Mouse Model of Subcutaneous Tumors

The protocol for the animal experiments was approved by the Animal Care and Use Committee of the National Institute of Radiological Sciences (code 07-1064-23, 25 September 2017), and all animal experiments were conducted following the institutional guidelines regarding animal care and handling. BALB/c-nu/nu male mice (5 weeks old, CLEA Japan, Tokyo, Japan) were maintained under specific pathogen-free conditions. Mice (*n* = 165) were inoculated subcutaneously with BxPC-3 cells (4 × 10^6^) in the left thigh under isoflurane anesthesia.

### 4.5. Immunohistochemistry with Anti-TNC Antibodies

When subcutaneous tumors reached a diameter of approximately 8 mm, tumors were irradiated with 30 Gy of X-rays at a rate of 3.9 Gy/min with a TITAN-320 X-ray generator (Shimadzu, Kyoto, Japan). Other parts of the mouse body were covered with a brass shield. On post-exposure days 1, 3, and 7, tumors (*n* = 3 per time point) were sampled and fixed in 10% (*v*/*v*) neutral buffered formalin and embedded in paraffin for sectioning. Nontreated tumors were used as controls. Sections (thickness, 1 μm) were immunostained with antibody 3–6 (diluted 1:200) followed by horseradish peroxidase-conjugated anti-rat immunoglobulin from a kit (BD, Franklin Lakes, NJ, USA). Nuclei were counterstained with hematoxylin.

To compare antibodies 3–6 and 12–2–7, six subcutaneous A375 tumors were fixed with 4% paraformaldehyde overnight in 0.1 M sodium phosphate (pH 7.2). After dehydration in ethanol, the tissues were embedded in polyester wax. Adjacent sections were immunostained with anti-TNC antibody 3–6 (6 μg/mL) or 12–2–7 (8 μg/mL) as the primary antibody; the secondary antibody was goat anti-rat IgG biotin conjugate (SC-2041, Santa Cruz Biotechnology, Dallas, TX, USA; diluted 1:1000). A streptavidin-peroxidase polymer (S2438, Sigma; diluted 1:1000) served as the detection reagent. Coloring was done with diaminobenzidine, and nuclei were stained with methyl green.

### 4.6. Radiolabeling of Antibodies

Antibodies were conjugated with *p*-SCN-Bn-CHX-A′′-DTPA (DTPA; Macrocyclics, Dallas, TX, USA) as previously described [16] and DTPA-conjugated antibodies were purified via centrifugation through columns of Sephadex G-50 (GE Healthcare, Little Chalfont, UK) (700× *g* for 2 min). For each antibody, the ratio of conjugated DTPA to antibody was ~1.5, as determined by cellulose acetate electrophoresis. Each DTPA-conjugated antibody (3–6, 16.3 μg; 12–2–7, 11.5 μg; TDEAR2, 15.3 μg; and control antibody, 27.8 μg) was mixed with 0.74 MBq (megabecquerel) ^111^InCl_3_ in 0.5 M acetate buffer (pH 6.0), and the mixture was incubated at room temperature for 30 min. The resulting radiolabeled antibodies were separated from free radionuclides via centrifugation through a Sephadex G-50 column, as noted above. The yield of radiolabeling ranged from 70.4% to 84.3%, and radiochemical purity exceeded 96%. The specific radioactivity was as follows: 3–6, 37.1 kBq/μg; 12–2–7, 50.8 kBq/μg; TDEAR2, 34.0 kBq/μg; and control antibody, 22.2 kBq/μg.

### 4.7. Biodistribution of ^111^In-Labeled Antibodies

When subcutaneous tumors reached a diameter of approximately 8 mm, tumors were irradiated with 0 or 30 Gy of X-rays. On post-exposure days 1, 3, and 7, mice (body weight, 22.1 ± 2.5 g) were intravenously injected with 37 kBq of an ^111^In-labeled antibody (3–6, 12–2–7, TDEAR2, and control antibody). The total injected protein dose was adjusted to 20 µg per mouse by adding the corresponding intact antibody. At 30 min post-injection, as well as on days 1, 2, and 4 post-injection, mice (*n* = 5 per time point) were euthanized by isoflurane inhalation, and blood was obtained from the heart. Tumors and major organs were removed and weighed, and radioactivity was measured using a gamma counter. The data are expressed as the percentage of injected dose per gram of tissue (% ID/g).

### 4.8. SPECT/CT with ^111^In-Labeled Antibodies

The BxPC-3 xenograft model mice (26.6 ± 0.5 g, *n* = 1 per group) were injected with approximately 1.85 MBq ^111^In-labeled antibody 3–6 or control antibody via a tail vein 7 days after irradiation with 0 or 30 Gy of X-rays. The injected antibody dose was adjusted to 50 μg per mouse by adding the corresponding intact antibody. At 30 min post-injection, as well as on days 1, 2, 3, and 4 post-injection, the mice were anesthetized with isoflurane and imaged with a SPECT/CT Preclinical Imaging system VECTor/CT equipped with a multi-pinhole collimator (MILabs, Utrecht, Netherlands). The SPECT scan time was 15 min for the 30 min and day 1 time points, 20 min for day 2, 25 min for day 3, and 30 min for day 4. SPECT images were reconstructed using a pixel-based ordered-subsets expectation-maximization algorithm with two subsets and eight iterations on a 0.8 mm voxel grid without correction for attenuation. CT data were acquired using an X-ray source set at a peak voltage of 60 kV and 615 μA after the SPECT scan, and the images were reconstructed using a filtered back-projection algorithm for the cone beam. Images were merged using PMOD software (ver. 3.4; PMOD Technology, Zürich, Switzerland).

### 4.9. Autoradiography

On day 1 post-injection of ^111^In-labeled antibody 3–6 or control antibody (1.85 MBq, 50 μg protein), tumors (*n* = 1 per group) were excised and frozen in Tissue-Tek O.T.C. compound (Sakura Finetek, Tokyo, Japan). Frozen sections (thickness, 20 μm) were fixed with 10% neutral buffered formalin, washed, and dried. The dried sections were exposed to an imaging plate (Fuji Film, Tokyo, Japan) and the imaging plate was scanned with an FLA-7000 image plate reader (Fuji Film). After reading, the sections were stained with hematoxylin and eosin (H&E). Signal intensity in six sections for each group was quantified by ImageJ (ver. 1.5.3, National Institutes of Health, Bethesda, MD, USA).

### 4.10. Statistical Analysis

Biodistribution data are expressed as the mean ± SD. The data were analyzed with two-way analysis of variance and the Sidak multiple comparison test using Prism 7 software (GraphPad Software, La Jolla, CA, USA). Signal intensity data are expressed as the mean ± SD and were analyzed with one-way ANOVA with the multiple comparison test using Prism. The criterion for statistical significance was *p* < 0.05.

## 5. Conclusions

Our anti-TNC antibody 3–6 labeled with ^111^In was aggressively taken up by tumors irradiated with X-rays compared with nonirradiated tumors and a control antibody. A drug delivery targeting a molecule, such as TNC, responding to antitumor therapy has the potential to provide better outcomes when combined with conventional therapy for refractory cancer.

## Figures and Tables

**Figure 1 cancers-12-03652-f001:**
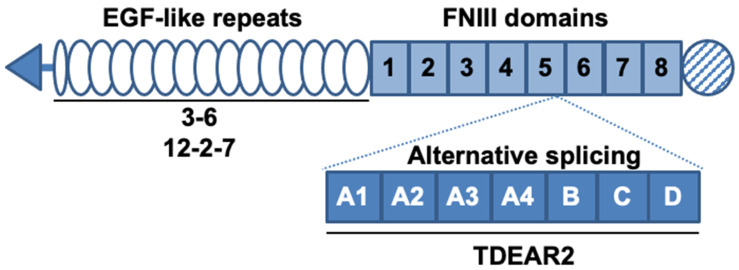
Schematic structure of tenascin-C (TNC) and the known binding sites of antibodies. TNC contains epidermal growth factor (EGF)-like repeats and fibronectin type III (FNIII) domains. Alternative splicing occurs between the fifth and sixth FNIII domains. The known binding sites of the three antibodies are denoted by solid lines under the domains. The antibodies 3–6 and 12–2–7 recognize the EGF-like repeats and TDEAR2 binds to the alternative splicing region.

**Figure 2 cancers-12-03652-f002:**
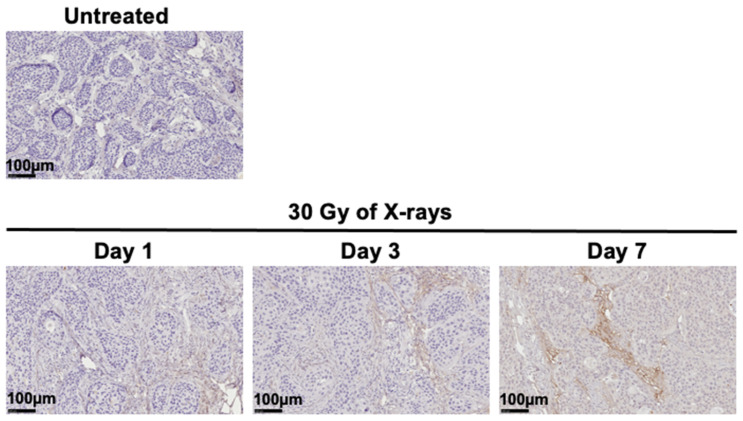
Immunohistochemical staining for TNC in BxPC-3 tumors. Paraffin-embedded sections were stained with anti-TNC antibody 3–6 (*n* = 3 per group). Shown are representative images of tumors that had been irradiated with X-rays (30 Gy) or not irradiated.

**Figure 3 cancers-12-03652-f003:**
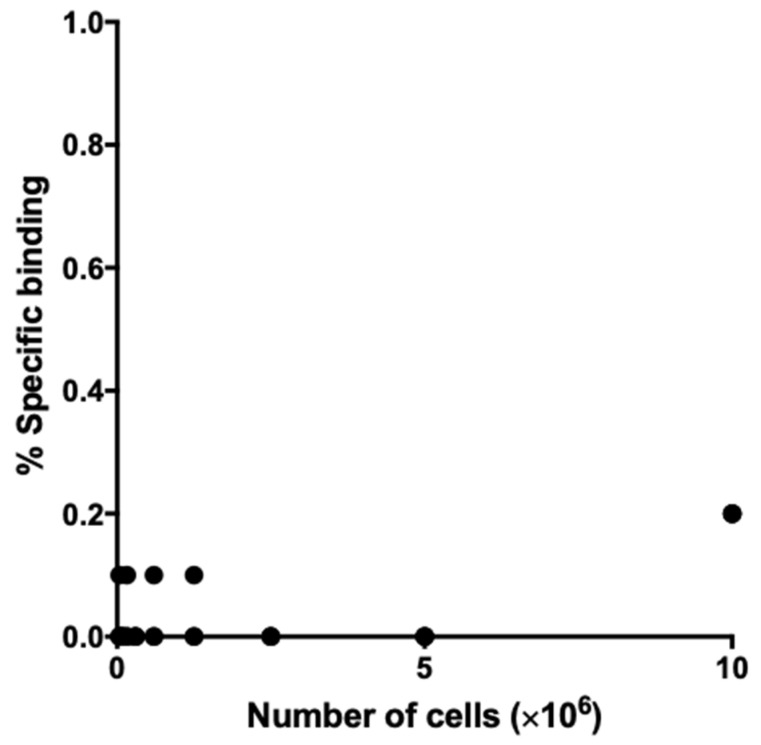
Binding of ^111^In-labeled anti-tenascin-C (TNC) antibody 3–6 to BxPC-3 cells (*n* = 3 per number of cells).

**Figure 4 cancers-12-03652-f004:**
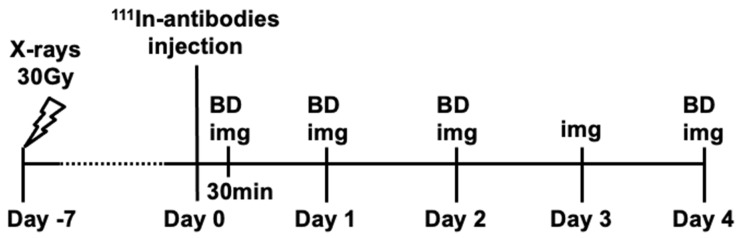
Schedule for the biodistribution and SPECT/CT (single-photon emission computed tomography with computed tomography) imaging studies. ^111^In-labeled antibodies were intravenously injected into mice bearing BxPC-3 tumors at day 7 after X-ray exposure (0 or 30 Gy). BD = biodistribution; Img = SPECT/CT imaging.

**Figure 5 cancers-12-03652-f005:**
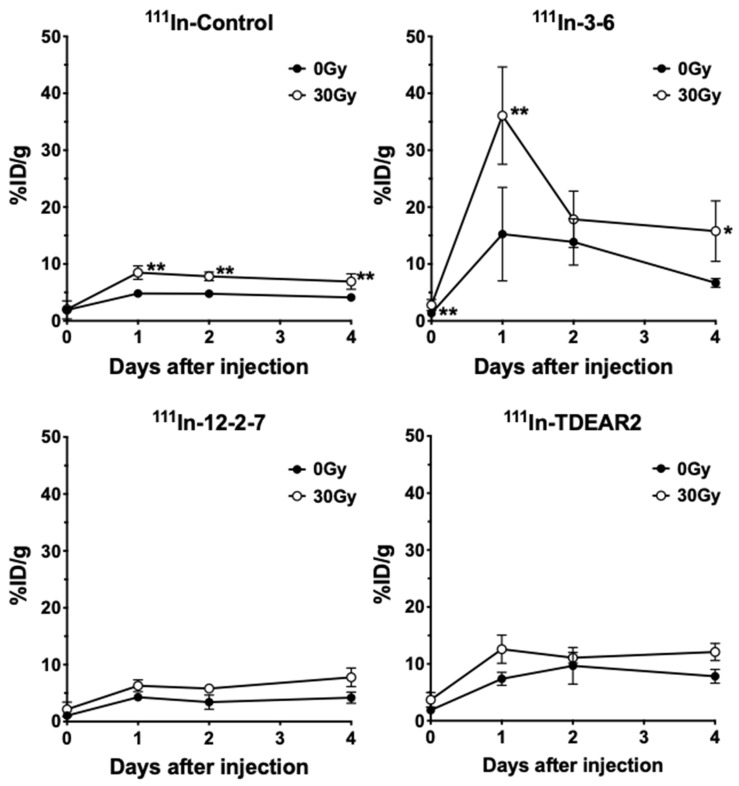
Time-activity curves of ^111^In-labeled antibodies in nonirradiated tumors (black circles) and tumors irradiated with 30 Gy (white circles) of X-rays. Mice (*n* = 5 per time point) were intravenously injected with 37 kBq (kilobecquerel) of an ^111^In-labeled anti-TNC antibody (3–6, 12–2–7, and TDEAR2) or a rat nonspecific antibody as a control. Biodistribution experiments were conducted after 30 min and on days 1, 2, and 4 post-injection. Data are expressed as the mean ± S.D. (*n* = 5). * *p* < 0.05, ** *p* < 0.01 (nonirradiated vs. 30 Gy). The uptakes in organs and tissues are shown in Table 1, Table 2, Table 3 and Table 4.

**Figure 6 cancers-12-03652-f006:**
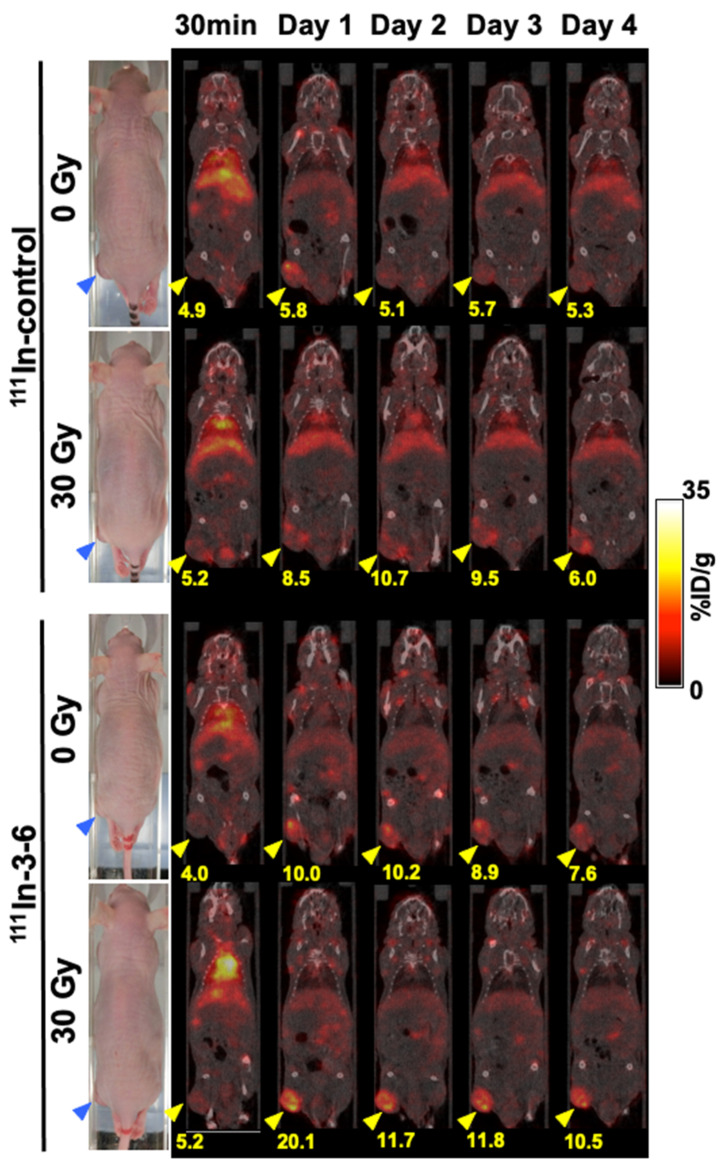
Coronal SPECT/CT images of nude mice bearing BxPC-3 tumors (*n* = 1 per group). The imaging study was conducted after 30 min and on days 1, 2, 3, and 4 after intravenous injection with 1.85 MBq ^111^In-labeled control antibody or antibody 3–6. Blue and yellow arrowheads indicate tumors. The numbers under tumors indicate the mean of % ID/g in the corresponding tumors.

**Figure 7 cancers-12-03652-f007:**
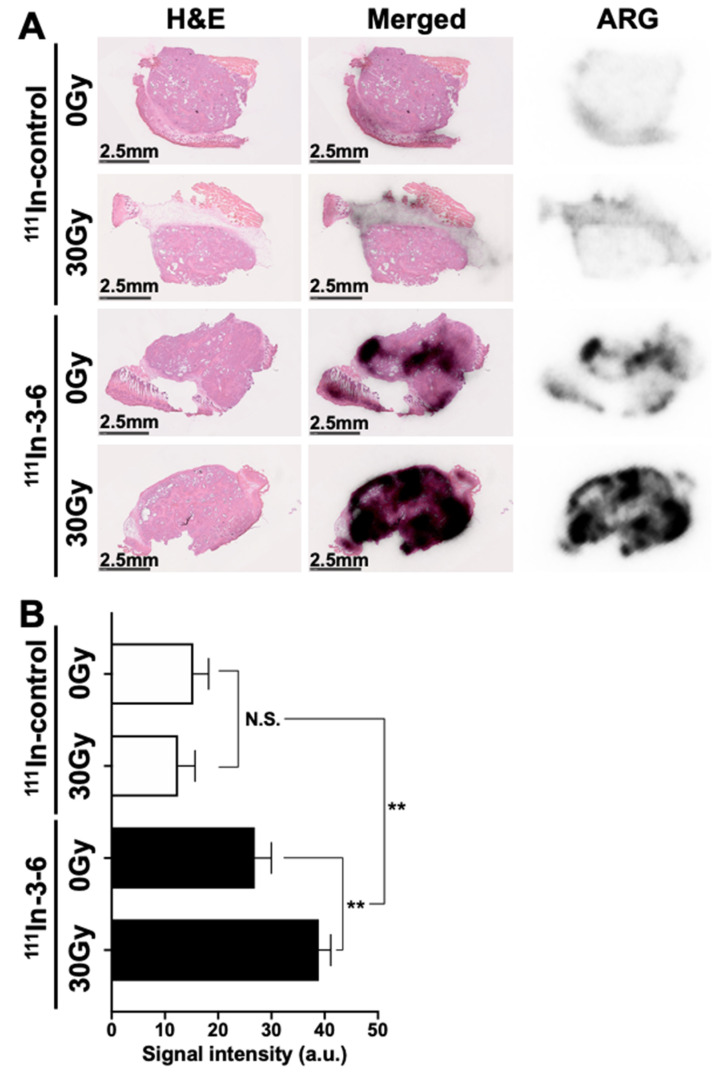
Autoradiography of ^111^In-labeled antibodies and H&E (hematoxylin and eosin) staining of BxPC-3 tumors (six sections per group). Tumors that were not irradiated or irradiated with 30 Gy of X-rays were sampled on day 1 post-injection with 1.85 MBq ^111^In-labeled control antibody or antibody 3–6. (**A**) Representative H&E-stained sections (left panels), merged images (center panels), and autoradiographic images (ARG, right panels). Scale Bar: 2.5 mm. (**B**) Quantitative analysis of autoradiographic images. ** *p* < 0.01; N.S., not significant., a.u., arbitrary unit.

**Figure 8 cancers-12-03652-f008:**
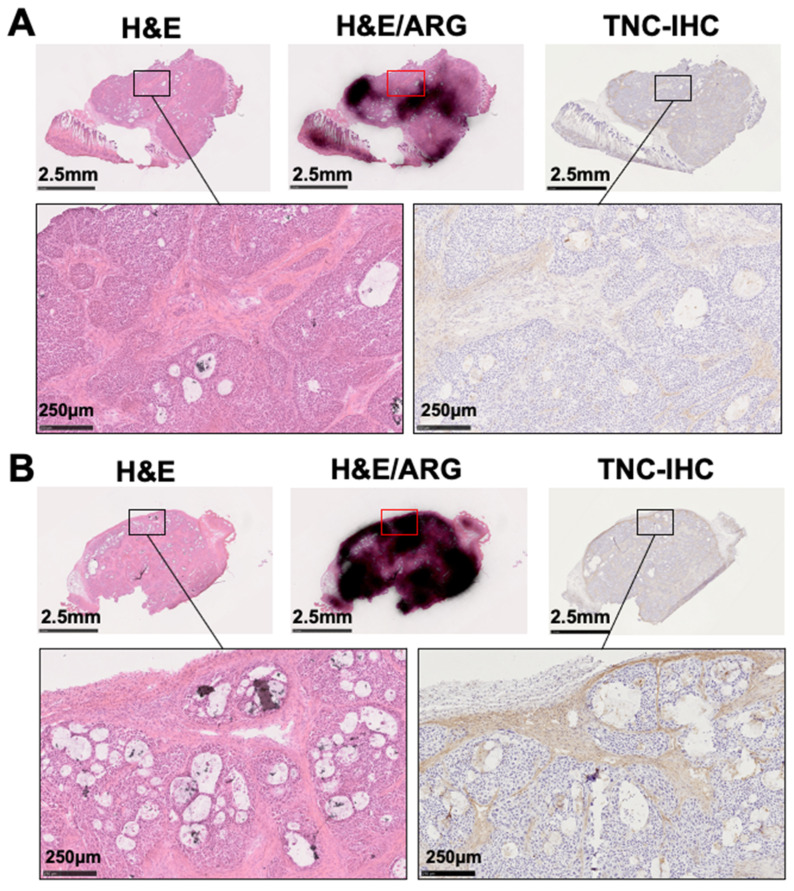
TNC-immunohistochemical stained sections adjacent to autographic images. Autoradiography of BxPC-3 tumor sections from mice that had been injected with ^111^In-labeled anti-TNC antibody 3–6, with subsequent staining with H&E. The adjacent sections were immunostained with anti-TNC antibody 3–6. The autographic and H&E images are the same as those shown in Figure 7. Nonirradiated tumors (**A**) and those irradiated with 30 Gy (**B**) of X-rays were sampled at day 1 post-injection with 1.85 MBq of ^111^In-labeled antibody 3–6.

**Figure 9 cancers-12-03652-f009:**
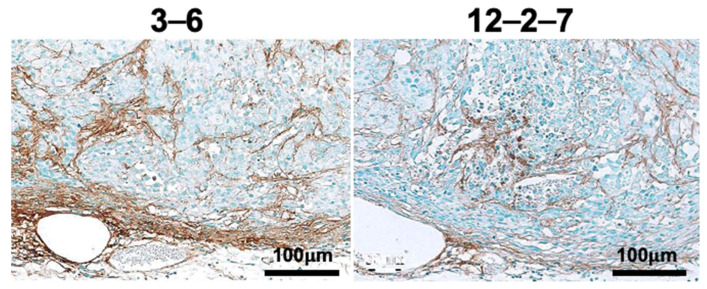
Immunohistochemical staining for TNC with antibody 3–6 (**left**) or 12–2–7 (**right**) in adjacent sections. Shown are representative images (*n* = 6 per antibody).

**Table 1 cancers-12-03652-t001:** Biodistribution of ^111^In-labeled control antibody in mice bearing BxPC-3 tumors.

Tissue/Organ	30 Min	Day 1	Day 2	Day 4
0 Gy				
Blood	40.6 ± 3.1	17.8 ± 1.3	15.7 ± 1.9	13.3 ± 1.3
Lung	13.8 ± 2.8	7.8 ± 0.6	6.6 ± 1.3	5.9 ± 1.1
Liver	12.1 ± 0.9	9.7 ± 1.1	8.7 ± 0.8	8.1 ± 1.2
Spleen	5.7 ± 1.3	3.8 ± 0.2	4.1 ± 0.5	4.4 ± 0.4
Pancreas	2.4 ± 0.2	1.9 ± 0.4	1.8 ± 0.5	1.6 ± 0.2
Intestine	2.4 ± 0.6	1.6 ± 0.3	1.5 ± 0.1	1.4 ± 0.2
Kidney	10.0 ± 1.7	6.8 ± 0.8	6.5 ± 0.3	5.7 ± 0.5
Muscle	0.7 ± 0.2	1.1 ± 0.1	1.1 ± 0.1	1.1 ± 0.1
Bone	3.0 ± 0.6	1.9 ± 0.2	1.9 ± 0.1	2.0 ± 0.3
30 Gy				
Blood	43.6 ± 0.9	20.9 ± 0.8	17.2 ± 1.5	13.4 ± 2.5
Lung	12.8 ± 1.5	9.5 ± 0.5	6.4 ± 0.7	6.5 ± 1.5
Liver	14.3 ± 1.4 **	9.2 ± 0.8	7.9 ± 1.0	6.3 ± 0.7 *
Spleen	6.3 ± 0.6	4.2 ± 0.7	4.3 ± 0.6	3.9 ± 0.9
Pancreas	2.4 ± 0.6	1.7 ± 0.2	1.6 ± 0.2	1.3 ± 0.1
Intestine	2.4 ± 0.3	1.4 ± 0.1	1.6 ± 0.2	1.4 ± 0.2
Kidney	10.2 ± 0.6	6.9 ± 0.4	6.3 ± 1.0	4.9 ± 0.6
Muscle	0.6 ± 0.1	1.3 ± 0.1	1.2 ± 0.2	1.1 ± 0.2
Bone	3.8 ± 0.4 *	2.6 ± 0.4 *	2.0 ± 0.4	2.0 ± 0.3

Data are expressed as the mean ± SD of % ID/g. * *p* < 0.05, ** *p* < 0.01, when compared with the 0 Gy group counterpart.

**Table 2 cancers-12-03652-t002:** Biodistribution of ^111^In-labeled anti-TNC antibody 3–6 in mice bearing BxPC-3 tumors.

Tissue/Organ	30 Min	Day 1	Day 2	Day 4
0 Gy				
Blood	38.9 ± 1.8	7.7 ± 0.3	6.9 ± 1.2	2.4 ± 0.8
Lung	12.8 ± 1.8	3.5 ± 0.3	3.2 ± 0.3	1.4 ± 0.4
Liver	8.2 ± 0.6	7.3 ± 0.6	6.8 ± 0.2	4.9 ± 0.9
Spleen	7.5 ± 0.7	9.0 ± 3.8	6.6 ± 0.3	3.8 ± 1.7
Pancreas	2.0 ± 0.4	0.8 ± 0.1	0.8 ± 0.1	0.3 ± 0.1
Intestine	3.7 ± 0.4	7.0 ± 1.1	5.2 ± 0.5	2.3 ± 0.5
Kidney	8.4 ± 0.4	4.2 ± 0.4	3.6 ± 0.3	2.3 ± 0.4
Muscle	0.6 ± 0.1	0.9 ± 0.0	0.9 ± 0.0	0.5 ± 0.1
Bone	4.3 ± 0.8	10.8 ± 1.5	11.1 ± 0.6	6.6 ± 0.7
30 Gy				
Blood	36.1 ± 4.9	7.7 ± 0.9	5.8 ± 0.8	3.5 ± 0.5
Lung	14.3 ± 2.4	3.4 ± 0.7	2.6 ± 0.3	1.8 ± 0.3
Liver	8.2 ± 0.6	7.8 ± 1.0	7.7 ± 0.6	7.3 ± 0.4 **
Spleen	9.2 ± 2.7	10.8 ± 3.8	9.9 ± 2.6	7.2 ± 0.6
Pancreas	1.9 ± 0.3	0.8 ± 0.2	0.7 ± 0.1	0.5 ± 0.1
Intestine	3.4 ± 0.7	6.6 ± 0.9	4.6 ± 1.0	3.1 ± 0.2
Kidney	8.8 ± 1.0	3.9 ± 0.4	3.6 ± 0.4	2.9 ± 0.5
Muscle	0.8 ± 0.2	1.0 ± 0.1	1.0 ± 0.3	0.6 ± 0.1
Bone	5.9 ± 1.3	11.5 ± 2.3	10.6 ± 1.2	8.3 ± 0.7

Data are expressed as the mean ± SD of % ID/g, ** *p* < 0.01, when compared with the 0 Gy group counterpart.

**Table 3 cancers-12-03652-t003:** Biodistribution of ^111^In-labeled anti-TNC antibody 12–2–7 in mice bearing BxPC-3 tumors.

Tissue/Organ	30 Min	Day 1	Day 2	Day 4
0 Gy				
Blood	33.5 ± 1.3	16.2 ± 2.1	14.7 ± 1.0	12.5 ± 0.8
Lung	13.0 ± 1.3	6.9 ± 0.7	5.7 ± 1.0	5.5 ± 1.1
Liver	6.3 ± 0.7	4.8 ± 0.7	4.4 ± 0.5	4.8 ± 0.3
Spleen	4.4 ± 0.5	3.2 ± 0.5	4.1 ± 0.8	3.7 ± 0.7
Pancreas	2.0 ± 0.5	1.5 ± 0.1	1.3 ± 0.2	1.3 ± 0.2
Intestine	1.8 ± 0.3	1.5 ± 0.2	1.4 ± 0.1	1.3 ± 0.1
Kidney	7.5 ± 0.6	5.6 ± 0.5	6.0 ± 0.3	5.0 ± 0.6
Muscle	0.6 ± 0.1	1.0 ± 0.1	1.1 ± 0.1	0.9 ± 0.1
Bone	2.7 ± 0.3	1.9 ± 0.1	2.1 ± 0.5	2.6 ± 0.2
30 Gy				
Blood	33.8 ± 1.5	18.3 ± 0.7	14.8 ± 0.5	11.5 ± 1.8
Lung	12.4 ± 1.8	6.3 ± 1.2	5.5 ± 0.8	5.6 ± 0.7
Liver	5.7 ± 0.5	4.1 ± 0.7	3.6 ± 0.4	4.5 ± 1.1
Spleen	4.1 ± 0.8	3.7 ± 0.7	3.3 ± 0.3	3.2 ± 0.4
Pancreas	1.4 ± 0.2 **	1.4 ± 0.2	1.4 ± 0.2	1.2 ± 0.2
Intestine	1.9 ± 0.4	1.3 ± 0.2	1.4 ± 0.2	1.2 ± 0.2
Kidney	7.5 ± 0.6	5.3 ± 0.9	4.7 ± 0.2 **	4.0 ± 0.3 *
Muscle	0.6 ± 0.1	1.0 ± 0.1	1.0 ± 0.1	0.9 ± 0.2
Bone	2.4 ± 0.1	2.2 ± 0.3	2.4 ± 0.1	2.3 ± 0.3

Data are expressed as the mean ± SD of % ID/g. * *p* < 0.05, ** *p* < 0.01, when compared with the 0 Gy group counterpart.

**Table 4 cancers-12-03652-t004:** Biodistribution of ^111^In-labeled anti-TNC antibody TDEAR2 in mice bearing BxPC-3 tumors.

Tissue/Organ	30 min	Day 1	Day 2	Day 4
0 Gy				
Blood	39.0 ± 1.8	17.7 ± 1.4	15.4 ± 1.0	12.6 ± 1.7
Lung	13.6 ± 1.5	7.9 ± 1.0	7.1 ± 0.8	5.9 ± 0.5
Liver	10.2 ± 0.4	5.6 ± 0.5	5.3 ± 0.8	4.9 ± 0.9
Spleen	10.2 ± 1.1	6.6 ± 1.6	6.0 ± 0.7	4.8 ± 1.1
Pancreas	2.0 ± 0.4	1.8 ± 0.2	1.7 ± 0.2	1.5 ± 0.3
Intestine	2.4 ± 0.2	2.2 ± 0.3	2.0 ± 0.3	1.7 ± 0.4
Kidney	11.1 ± 0.6	6.4 ± 1.1	6.5 ± 0.7	5.1 ± 0.8
Muscle	0.9 ± 0.2	1.4 ± 0.3	1.2 ± 0.2	1.1 ± 0.1
Bone	3.8 ± 0.5	4.5 ± 0.6	4.9 ± 0.9	4.5 ± 0.6
30 Gy				
Blood	40.8 ± 2.0	19.7 ± 1.2	16.8 ± 0.7	12.3 ± 1.0
Lung	15.3 ± 2.6	8.4 ± 0.7	7.0 ± 0.6	6.6 ± 1.9
Liver	9.3 ± 0.7 *	6.2 ± 0.1 **	5.1 ± 0.5	5.1 ± 0.6
Spleen	13.2 ± 1.8	9.3 ± 2.2	7.8 ± 0.7	5.4 ± 1.3
Pancreas	1.9 ± 0.6	1.7 ± 0.2	1.4 ± 0.1	1.3 ± 0.1
Intestine	2.1 ± 0.3	2.4 ± 0.2	2.1 ± 0.3	1.5 ± 0.2
Kidney	10.1 ± 0.8	6.2 ± 1.3	5.6 ± 0.4	4.9 ± 0.5
Muscle	0.8 ± 0.2	1.3 ± 0.2	1.1 ± 0.2	1.0 ± 0.1
Bone	4.0 ± 0.4	4.8 ± 0.4	4.6 ± 0.6	3.7 ± 0.7

Data are expressed as the mean ± SD of % ID/g. * *p* < 0.05, ** *p* < 0.01, when compared with the 0 Gy group counterpart.

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
