# Peer review of "Proof of Concept Study for Increasing Tenascin-C-Targeted Drug Delivery to Tumors Previously Subjected to Therapy: X-Irradiation Increases Tumor Uptake"

_cancers, 2020, doi:10.3390/cancers12123652_

Round 1
Reviewer 1 Report
This is a review of the manuscript untitled Proof of Concept Study for Increasing Tenascin-C targeted Drug Delivery to Tumors Previously Subjected to Therapy: X-Irradiation Increases Tumor Uptake, submitted to the journal Cancers. The manuscript number is cancers-989064. This manuscript aims to elaborate a proof of concept that an antibody recognizing Tenascin C (specific factor), which is involved in tissue injury repair, could achieve the goal of delivering an additional radionuclide (as antitumor agent) to tumors during tissue repair after initial anticancer therapy. The authors demonstrated a markedly increased tumor uptake after radiation exposure in a pancreatic cancer mouse model. This finding strongly supports their concept. Their proposed therapeutic strategy could result in better outcomes for patients with treatment-refractory cancer, including pancreatic cancers, associated with poor prognosis.
This manuscript is very well written: summary, abstract, introduction, overall presentation including discussion and material and methods.
I recommend accepting this manuscript in present form. This manuscript is up to the higher standard of the journal, presenting a very interesting proof of concept with an impressive amount of work, aiming to improve the outcomes of patients with refractory (pancreatic) cancers.
Reviewer 2 Report
I recommend the authors to continue their research to address the dubious aspects that they themselves have highlighted
Reviewer 3 Report
The authors have addressed my comments and I have no additional concerns.
Reviewer 4 Report
The authors convinced me in that the m/s is a kind of prerequisite for a future success with the trials of their approach with the really cytocidal nuclide. Therefore I think that the manuscript can be published in Cancers.
This manuscript is a resubmission of an earlier submission. The following is a list of the peer review reports and author responses from that submission.
Round 1
Reviewer 1 Report
This is a review of the manuscript untitled Proof of Concept Study for Increasing Tenascin-C targeted Drug Delivery to Tumors Previously Subjected to Therapy: X-Irradiation Increases Tumor Uptake, submitted to the journal Cancers. The manuscript number is cancers-989064. This manuscript aims to elaborate a proof of concept that an antibody recognizing Tenascin C (specific factor), which is involved in tissue injury repair, could achieve the goal of delivering an additional radionuclide (as antitumor agent) to tumors during tissue repair after initial anticancer therapy. The authors demonstrated a markedly increased tumor uptake after radiation exposure in a pancreatic cancer mouse model. This finding strongly supports their concept. Their proposed therapeutic strategy could result in better outcomes for patients with treatment-refractory cancer, including pancreatic cancers, associated with poor prognosis.
This manuscript is very well written: summary, abstract, introduction, overall presentation including discussion and material and methods.
I recommend accepting this manuscript in present form. This manuscript is up to the higher standard of the journal, presenting a very interesting proof of concept with an impressive amount of work, aiming to improve the outcomes of patients with refractory (pancreatic) cancers.
Reviewer 2 Report
The authors hypothesize that an agent recognizing a molecule that respond to anticancer treatment-induce tissue injury could deliver an additional antitumor agent including a radionuclide to damaged cancer tissues during repair.
The working hypothesis is interesting, but the experimental design is not clearly exposed in the materials and methods.
Moreover, the authors themselves write that their study has numerous limitations. There are still several points to investigate before being able to publish the work, as the authors themselves write: 1) the stroma of non-irradiated BxPC-3 tumours expressed only a small amount of TNC; 2) upregulation of TNC expression is induced by both anticancer drugs and radiation. I believe that these dubious aspects should be better investigated before publishing the work. The conclusions may not turn out to be true. The working hypothesis must be supported by more certain data.
In particular:
Section 4.4: How many animals were used?
The manuscript paragraph 2.3 figure 6: 30 min and 1 day are described, but 2, 3 and 4 days are missing.
Tables (1-4) are not sufficiently described in the text.
Reviewer 3 Report
The authors are interested in whether a molecule involved in a tumor’s response to treatment (in this case X-irradiation) can be used to deliver anticancer drugs or radionuclides. They use the BxPC-3 pancreatic cancer xenograft tumor model in which they irradiate (or not) and they selected Tenascin-C (TNC) as their molecule to target. First they compared TNC staining in BxPC-3 control tumors and irradiated tumors and found that TNC staining is very weak in control tumors, but increases fairly significantly by 7 days post irradiation. They next tested the bio-distribution of radiolabeled control antibodies or 3 antibodies to TNC. They found that 1 of the TNC antibodies (3-6) they tested showed a significant increase in uptake in the irradiated tumors compared to the non-irradiated tumors and another (12-2-7) showed a modest but statistically significant increase. The third did not show any increase, which they hypothesize is because it recognizes a splice variant of TNC and perhaps this splice variant is not present n the tumors. They also looked at the bio-distribution of the antibodies in several non-tumor tissues in the tumor bearing mice and for the most part did not see any major differences between irradiated and non-irradiated mice. Next they did single-photon emission computed tomography and computed tomography (SPECT/CT) imaging of the mice injected with labeled control or TNC antibodies with and without irradiation. They saw that the labeled antibody targeting TNC accumulated in the tumors much more than the control antibody following irradiation. They sectioned the tumors and used autoradiography to detect antibody signal and saw significant signal in the irradiated tumors from mice injected with the TNC antibody. They did TNC staining in adjacent sections and saw strong stromal staining in areas that also showed strong autoradiography signal.
The paper explores a very interesting hypothesis that would be of interest to the field. The paper is clearly written and the presented data are consistent with their conclusions. However, I have major concerns about the rigor of the studies and some of their conclusions that need to be addressed:
Major concerns:
- Several figures seem to show a single tumor for each treatment type. Figure 5 indicates that n=5 mice, but figures 6-9 only show a signal mouse. Analysis of one mouse is not sufficient to make any conclusions. We need to see the data for all the mice. I assume the figures show representative images from one mouse out of several. If so, they need to show the data for the other mice. Ideally they would add a graph showing the quantitation of the signal from each mouse in the various analyses they did, and a statistical analysis would be done. This would hopefully demonstrate that the differences they clearly observe in the 1 representative mouse are reproducible and statistically significant.
- Figure 8 lacks important controls. They should stain for TNC in the non-irradiated mice that showed weak signal. Though they show weak TNC signal in non-irradiated tumors in figure 2, it is important here because they are using this figure to demonstrate that the strong signal they observe in Figures 6 and 7 is strongly correlated with increased TNC in the same regions. We would expect to see weak to no staining for TNC in the non-irradiated mice with low SPECT/CT and autoradiography signal.
- Statistics are missing from most of the figures raising concerns about how rigorously this work was done.
Minor comments:
- The Figure legends do not have enough detail to properly interpret the experiments. More detail would be helpful.
- Figure 3 is hard to interpret, what does positive binding look like? How many times was this done? No statistics are provided.
- While overall I agree with their conclusions (assuming they can provide the above requested data showing reproducibility and significance). However, they make some claims in the discussion that seem a bit overstated. These are easily fixed by rephrasing their statements and/or pointing out when more data is needed to confirm this suggestion. Here re some of the statements that concern me:
- Lines 188-192: “Our proposed therapeutic strategy with the …… molecular-weight agents, into tumors [9]. The authors seem to be suggesting that their data shows this, but they are making several assumptions here that they did not prove or support by citing other studies. They should revise this to make it clear which statements are supported (by other studies) and which are suggestions they make from their work that would require additional work to prove.
- Lines 207-209: “To our knowledge,…. complexity of TNC functions during tissue repair.” Here they seem to suggest that their work describes the splicing of TNC pre-mRNA in tumors, but they did not show this, their data suggests it may be the case since the antibody that does not give signal recognizes a splice variant, but this alone does not prove splicing exists. They should revise this statement to indicate their data suggests that perhaps…., and to date no other studies have shown this.
Reviewer 4 Report
The m/s opens a wide avenue for the search for newer tumor cell-recognizing and drug-delivering substances. The authors succeeded to find the antibody recognizing particular domain in tenascin molecule essential for the protein’s function in tissue repair. The study is timely and well-designed. Although the authors present the study as a proof of principle some additional experiments may be useful to support the idea of the author’s approach. First, it is unclear can the curve of In-111-labeled 3-6 antibody uptake with the sharp peak at Day 1 and then strong decline reflect any possible therapeutic effect. Therefore, the authors are requested to analyze the amount of cells dying from radioactivity in their model of therapy; the parameter should be measured in a few (7-12) days after delivery of 3-6 antibody-radioactive factor complex. Additionally, since extracellular matrix supports the growth of both cancer and stromal cells as shown on all sections stained with 3-6 antibody (especially Fig.9), it seems important to analyze the death of stromal cells in tumor area. Secondly, to further prove the concept and its possible clinical application I recommend the authors to substitute the radioactive agent by another cytotoxic factor, for instance docetaxel or gemcitabine. Generally, I suggest that the paper besides the quite interesting idea should give the readers more data convincing its feasibility and so the authors are requested to perform additional experiments.